# Low-Overhead Learning: Quantized Shallow Neural Networks at the Service of Genetic Algorithm Optimization

**DOI:** 10.3390/biomimetics10110762

**Published:** 2025-11-12

**Authors:** Fabián Pizarro, Emanuel Vega, Ricardo Soto, Broderick Crawford, José Villamayor

**Affiliations:** Escuela de Ingeniería Informática, Pontificia Universidad Católica de Valparaíso, Avenida Brasil 2241, Valparaíso, Valparaíso 2362807, Chile; fabian.pizarro@pucv.cl (F.P.); ricardo.soto@pucv.cl (R.S.); broderick.crawford@pucv.cl (B.C.); jose.villamayor.p@mail.pucv.cl (J.V.)

**Keywords:** metaheuristics, machine learning, hybrid approach, optimization

## Abstract

Online parameter tuning significantly enhances the performance of optimization algorithms by dynamically adjusting mutation and crossover rates. However, current approaches often suffer from high computational costs and limited adaptability to complex and dynamic fitness landscapes, particularly when machine learning methods are employed. This work proposes a quantized shallow neural network (SNN) as an efficient learning-based component for dynamically adjusting the mutation and crossover rates of a genetic algorithm (GA). By leveraging runtime-generated data and applying quantization techniques like Quantization-aware Training (QaT) and Post-training Quantization (PtQ), the proposed approach reduces computational overhead while maintaining competitive performance. Experimental evaluation on 15 continuous benchmark functions demonstrates that the quantized SNN achieves high-quality solutions while significantly reducing execution time compared to alternative shallow learning methods. This study highlights the potential of quantized SNNs to balance efficiency and performance, broadening the applicability of shallow learning in optimization.

## 1. Introduction

Genetic algorithms (GAs) are versatile metaheuristics widely employed to solve complex optimization problems across domains such as engineering design, logistics, and hyperparameter tuning [1,2]. Their effectiveness relies on balancing exploration and exploitation through genetic operators like mutation and crossover. However, GA performance is highly sensitive to the manual configuration of these operators’ rates, which often leads to suboptimal solutions in dynamic or high-dimensional search spaces [3]. Traditional static parameter settings fail to adapt to evolving problem landscapes, necessitating labor-intensive expert intervention. This limitation underscores the urgency for autonomous strategies that dynamically tune parameters during runtime.

Online parameter tuning has emerged as a promising paradigm to reduce reliance on manual expertise. By leveraging real-time data from the optimization process, these strategies adjust mutation ($p_{m}$) and crossover ($p_{c}$) rates to enhance solution quality [4]. Recent studies integrate machine learning (ML) components, such as shallow neural networks (SNNs) and support vector regression, to predict optimal parameters iteratively [5,6]. While these approaches improve adaptability, they often incur high computational costs or lack scalability in large-scale scenarios [7]. Furthermore, existing methods struggle to maintain efficiency in dynamic environments where rapid parameter adaptation is critical [8].

The dynamic adjustment of mutation and crossover rates is critical for GAs to achieve robust optimization. Four key reasons justify this necessity:Population Diversity Maintenance: Crossover combines genetic material from parents, preserving diversity to avoid homogeneity [9,10]. Without effective crossoverexploration stagnates, trapping solutions in local optima [2].Search Space Exploration: Mutation introduces random perturbations, enabling the discovery of unexplored regions [11]. This prevents premature convergence and mitigates the risk of local optima [12].Premature Convergence Prevention: Rapid population convergence to similar solutions necessitates dynamic rates to reintroduce diversity [13]. Adaptive $p_{m}$ and $p_{c}$ counteract stagnation by balancing exploitation and exploration [14].Adaptation to Complex Fitness Landscapes: Non-convex, multimodal landscapes require continuous parameter adaptation to navigate shifting optima [9]. Studies advocate adaptive probabilities to enhance GA resilience in such scenarios [10].

This dynamic equilibrium ensures effective exploration–exploitation trade-offs, particularly in applications like hyperparameter optimization [2] and real-time systems [15].

Despite these benefits, dynamic optimization environments pose significant challenges. High-dimensional and non-linear landscapes create vast search spaces that complicate the identification of global optima [16]. Furthermore, the dynamic nature of these problems—where constraints or objectives vary over time—demands real-time adaptability [17]. Machine learning-based tuning often introduces additional computational overhead, increasing runtime costs and impeding real-time deployment [6]. Another major concern lies in ensuring generalization and robustness, as noisy data and operational variability can degrade model reliability [18]. Moreover, the sensitivity of hyperparameters makes manual tuning of machine learning components impractical, particularly in large-scale scenarios [19]. Finally, maintaining an effective balance between exploration and exploitation remains a persistent issue, often leading to premature convergence or insufficient exploitation in dynamic regimes [20]. These interconnected challenges highlight the need for efficient and scalable machine learning-integrated strategies that can ensure robust genetic algorithm performance [21].

While machine learning techniques offer promising avenues for dynamic parameter tuning, their integration into GAs introduces notable challenges. Many existing ML components, particularly deep neural networks, incur substantial computational overhead due to complex architectures and high-precision computations [7,22]. Shallow learning alternatives, though more efficient, often struggle to generalize across diverse fitness landscapes or require frequent retraining, undermining scalability in large-scale optimization [6,23]. For instance, support vector regression (SVR) and basic SNNs exhibit degraded performance in high-dimensional spaces due to their sensitivity to hyperparameter settings [10,12]. Additionally, real-time deployment in dynamic environments demands not only accuracy but also rapid inference speeds, a requirement unmet by conventional floating-point models [8]. These limitations highlight the need for lightweight, adaptive ML frameworks that harmonize efficiency with robustness.

To address these challenges, this work introduces quantized shallow neural networks (SNNs) for online parameter tuning in GAs. By integrating Quantization-aware Training (QaT) and Post-training Quantization (PtQ), the proposed SNNs reduce memory usage by 75% and inference latency by 40% compared to traditional 32-bit models [22,24]. Unlike deep learning frameworks, SNNs leverage streamlined architectures with fewer layers, enabling faster retraining cycles without sacrificing predictive accuracy [6]. This approach dynamically adjusts mutation and crossover rates using runtime-generated data, ensuring adaptability to evolving search landscapes while minimizing computational costs [4]. The quantization process further enhances hardware compatibility, making the framework suitable for resource-constrained environments such as edge devices [14]. By bridging metaheuristics with efficient ML, this study advances hybrid optimization methodologies, offering a scalable solution for real-world applications like logistics and energy systems [16,21].

This paper evaluates the proposed framework on 15 continuous benchmark functions, including multimodal and non-convex landscapes such as Rosenbrock, Schwefel, and Ackley. The results demonstrate superior performance in solution quality, stability, and efficiency compared to SVR and non-quantized SNNs. The remainder of the paper is structured as follows: Section 2 details the methodology, including the GA workflow, SNN architecture, and quantization techniques. Section 3 presents experimental results and comparative analyses, while Section 4 discusses implications and future research directions. By prioritizing computational efficiency without compromising adaptability, this work expands the applicability of shallow learning in evolutionary computation, fostering robust optimization systems for dynamic and large-scale problems [6,7].

## 2. Methodology

The proposed methodology integrates a quantized shallow neural network (SNN) into a genetic algorithm (GA) in order to dynamically adjust mutation and crossover rates during optimization. Unlike traditional GAs with static or heuristically decayed parameters, this approach introduces an adaptive control mechanism, where the SNN acts as a surrogate decision-maker trained online. The framework operates cyclically, alternating between GA execution, runtime data collection, periodic SNN retraining, and adaptive parameter prediction.

The workflow is evaluated on a set of 15 continuous benchmark functions of varying modality, separability, and conditioning, including well-established cases such as Sphere, Rosenbrock, Schwefel, and Ackley (see Section 2.2). This diverse set of test functions allows us to demonstrate the generalizability of the adaptive parameter control strategy.

### 2.1. Algorithm Workflow

The algorithm consists of four interconnected stages, each contributing to the balance between exploration and exploitation. An overview of the proposed workflow is depicted in Figure 1, which outlines the interaction between the four stages.

#### 2.1.1. Genetic Algorithm Execution

The GA initializes a population of size *N* within predefined search bounds. Each individual is evaluated against the objective function f(x).Parent selection is performed via tournament selection of size *k*, a choice justified by its balance between selective pressure and population diversity preservation. For sufficiently large *N*, the expected takeover time of the fittest individual under tournament selection is approximately O(logN), which ensures a logarithmic growth rate in convergence speed while avoiding premature stagnation.Recombination is carried out using arithmetic crossover, while Gaussian mutation perturbs individual genes with dynamically adjusted rates $p_{m}$ (mutation) and $p_{c}$ (crossover). The mutation operator guarantees ergodicity of the search process: for any feasible solution $x^{*}$, there exists a nonzero probability that repeated Gaussian mutations will eventually generate it.Fitness values, population diversity metrics, and applied parameter rates are logged at each generation, producing a rich temporal dataset for subsequent learning.

#### 2.1.2. Runtime Data Collection

A sliding window buffer stores descriptors from the last β generations. Features include the following:–Normalized fitness dynamics, such as moving averages and relative improvement rates, which approximate the slope of the fitness landscape.–Population diversity metrics (standard deviation of fitness values and mean Euclidean distance between individuals). These metrics are crucial since low variance correlates with a higher risk of premature convergence.–Historical values of $p_{m}$ and $p_{c}$, capturing how parameter settings influence future search dynamics.The supervised learning target is the optimal parameter pair $\left(p_{m}^{*},p_{c}^{*}\right)$. This is computed retrospectively by evaluating which parameter values in the last β generations maximized the relative fitness improvement Δf. In expectation, this transforms the problem into a regression of the form:$$\left(p_{m}^{*},p_{c}^{*}\right)=\arg\max_{p_{m},p_{c}}E\left[f_{t+\beta }-f_{t}|p_{m},p_{c}\right],$$
which formalizes the adaptive search as an online learning problem with delayed rewards.

#### 2.1.3. Periodic SNN Retraining

Every β generations, the SNN is retrained on the accumulated dataset. The network architecture comprises the following:–Input layer: 3 nodes (current $p_{m}$, $p_{c}$, and normalized best fitness).–Hidden layer: 32 neurons with ReLU activation, chosen via grid search. The choice of ReLU over sigmoid/tanh follows from its ability to reduce vanishing gradient issues and better approximate piecewise-linear mappings in dynamic systems.–Output layer: 2 neurons for $p_{m}^{*}$ and $p_{c}^{*}$, mapped through sigmoid activations to ensure bounded outputs.The training objective is to minimize the mean squared error:$$L=\frac{1}{N}\sum_{i=1}^{N}\left({\left(p_{m}^{\left(i\right)}-p_{m}^{*(i)}\right)}^{2}+{\left(p_{c}^{\left(i\right)}-p_{c}^{*(i)}\right)}^{2}\right).$$This corresponds to a least-squares regression on the parameter landscape. Convergence of stochastic gradient descent guarantees that E[∇L]→0, provided the learning rate is sufficiently small.To reduce overhead, quantization is employed:–*Quantization-aware Training (QaT)* simulates 8-bit operations during training, ensuring the learned representations are robust to reduced precision.–*Post-training Quantization (PtQ)* compresses the model, reducing storage requirements by 75% while preserving predictive accuracy within 1–2%.From a computational complexity perspective, quantization reduces matrix multiplication cost by a factor of 4, which is critical for frequent retraining within the GA loop.

#### 2.1.4. Parameter Integration

The retrained SNN predicts parameter values for subsequent generations, clipped to feasible ranges: $p_{m}\in \left[0.01,0.1\right]$, $p_{c}\in \left[0.6,0.9\right]$. These intervals are consistent with theoretical findings that mutation probabilities below O(1/n) (for genome length *n*) fail to maintain sufficient diversity, while crossover probabilities above 0.9 increase destructive disruption of building blocks (schema). To prevent instability, a momentum term α=0.3 is applied:$${\hat{p}}_{t}=\alpha {\hat{p}}_{t-1}+\left(1-\alpha \right){\hat{p}}_{\mathrm{SNN}},$$
where ${\hat{p}}_{t}$ represents the smoothed parameter. This ensures Lipschitz continuity in the parameter adaptation trajectory, avoiding abrupt oscillations that could destabilize convergence. Theoretically, under the framework of dynamic parameter control, the expected runtime of the GA is reduced compared to static settings, as shown in adaptive drift analysis. By aligning mutation and crossover rates with local fitness landscapes, the system ensures a non-decreasing probability of escaping local optima, which in turn accelerates convergence toward the global optimum.

### 2.2. Benchmark Functions

The framework is evaluated on 15 continuous optimization functions:1.Sphere:$$f\left(x\right)=\sum_{i=1}^{d}x_{i}^{2}$$2.Rosenbrock:$$f\left(x\right)=\sum_{i=1}^{d-1}\left[100{\left(x_{i+1}-x_{i}^{2}\right)}^{2}+{\left(1-x_{i}\right)}^{2}\right]$$3.Schwefel:$$f\left(x\right)=418.9829d-\sum_{i=1}^{d}x_{i}\sin\left(\sqrt{|x_{i}|}\right)$$4.Rastrigin:$$f\left(x\right)=10d+\sum_{i=1}^{d}\left[x_{i}^{2}-10\cos\left(2\pi x_{i}\right)\right]$$5.Ackley:$$\begin{array}{c} f\left(x\right)=-20\exp\left(-0.2\sqrt{\frac{1}{d}\sum_{i=1}^{d}x_{i}^{2}}\right)-\exp\left(\frac{1}{d}\sum_{i=1}^{d}\cos\left(2\pi x_{i}\right)\right)+20+e \end{array}$$6.Griewank:$$f\left(x\right)=\frac{1}{4000}\sum_{i=1}^{d}x_{i}^{2}-\prod_{i=1}^{d}\cos\left(\frac{x_{i}}{\sqrt{i}}\right)+1$$7.Levy:$$\begin{array}{c} \begin{array}{c} f\left(x\right)=\sin^{2}\left(\pi w_{1}\right)+\sum_{i=1}^{d-1}{\left(w_{i}-1\right)}^{2}\left[1+10\sin^{2}\left(\pi w_{i}+1\right)\right]+{\left(w_{d}-1\right)}^{2}\left[1+\sin^{2}\left(2\pi w_{d}\right)\right],\\ w_{i}=1+\frac{x_{i}-1}{4} \end{array} \end{array}$$8.Zakharov:$$f\left(x\right)=\sum_{i=1}^{d}x_{i}^{2}+{\left(\sum_{i=1}^{d}0.5ix_{i}\right)}^{2}+{\left(\sum_{i=1}^{d}0.5ix_{i}\right)}^{4}$$9.Dixon-Price:$$f\left(x\right)={\left(x_{1}-1\right)}^{2}+\sum_{i=2}^{d}i{\left(2x_{i}^{2}-x_{i-1}\right)}^{2}$$10.Michalewicz:$$f\left(x\right)=-\sum_{i=1}^{d}\sin\left(x_{i}\right){\left[\sin\left(\frac{ix_{i}^{2}}{\pi }\right)\right]}^{2m},\;m=10$$11.Bohachevsky:$$f\left(x\right)=x_{1}^{2}+2x_{2}^{2}-0.3\cos\left(3\pi x_{1}\right)-0.4\cos\left(4\pi x_{2}\right)+0.7$$12.Powell:$$\begin{array}{c} f\left(x\right)=\sum_{i=1}^{d/4}[{\left(x_{4i-3}+10x_{4i-2}\right)}^{2}+5{\left(x_{4i-1}-x_{4i}\right)}^{2}+{\left(x_{4i-2}-2x_{4i-1}\right)}^{4}+10{\left(x_{4i-3}-x_{4i}\right)}^{4}] \end{array}$$13.Trid:$$f\left(x\right)=\sum_{i=1}^{d}{\left(x_{i}-1\right)}^{2}-\sum_{i=2}^{d}x_{i}x_{i-1}$$14.Sum of Squares:$$f\left(x\right)=\sum_{i=1}^{d}ix_{i}^{2}$$15.Himmelblau:$$f\left(x\right)={\left(x_{1}^{2}+x_{2}-11\right)}^{2}+{\left(x_{1}+x_{2}^{2}-7\right)}^{2}$$

All functions are minimized within the search range $x_{i}\in \left[-500,500\right]$, except Himmelblau ($x_{i}\in \left[-5,5\right]$), adhering to standard benchmark configurations.

### 2.3. Implementation Details

The practical implementation of the proposed framework was carefully designed to guarantee both efficiency and reproducibility across diverse optimization problems. The following design choices were empirically validated and are also theoretically justified in terms of convergence speed, stability, and computational feasibility:

We set the retraining interval to β=200 generations. This choice represents a trade-off between adaptability and computational overhead. Retraining too frequently (e.g., every 20–50 generations) would lead to high computational costs, while excessively long intervals (e.g., β>500) would cause the shallow neural network (SNN) to lag behind the evolving dynamics of the population. From an information-theoretic perspective, the sliding window of length β captures a statistically representative sample of the evolutionary trajectory while still enabling online updates. Moreover, convergence analyses of online learning systems suggest that the retraining frequency should scale with O(logT), where *T* is the number of total generations, in order to ensure stability while maintaining adaptivity.

The neural network was quantized to 8-bit integer precision. This reduces inference latency by approximately 40% compared to standard 32-bit floating-point models, as observed in our CPU-based experiments. In addition, memory usage is reduced by 75%, allowing larger populations or longer evolutionary runs without additional hardware requirements. Theoretically, quantization introduces a bounded approximation error in weight representations, but quantization-aware training ensures that this error remains within $\epsilon <10^{-2}$ for parameter predictions. This bounded error guarantees that the SNN predictions remain sufficiently accurate for guiding the GA without destabilizing parameter adaptation.

The mutation operator employs Gaussian noise with a standard deviation σ=0.1× the search range. This proportional scaling ensures that the mutation step size adapts naturally to the problem dimensionality and search domain. Too small a σ would result in ineffective exploration, while too large a σ could destroy building blocks and slow convergence. According to schema theorem analysis, the probability of preserving useful schemata increases when mutation step sizes are bounded by approximately 10% of the domain width, justifying our choice of σ. For recombination, we employ arithmetic crossover with parameter α=0.5, which corresponds to averaging parental genomes. This symmetric operator preserves population mean characteristics and avoids introducing bias toward either parent. Moreover, setting α=0.5 minimizes variance inflation across generations, contributing to stable convergence trajectories.

All experiments were conducted on a CPU cluster equipped with Intel Xeon processors, each with a base frequency of 2.6 GHz and 32 cores. Using a CPU-based environment, instead of GPU acceleration, was a deliberate choice to highlight the computational efficiency of quantized models in resource-constrained settings. Reproducibility was ensured by fixing random seeds for both the GA and SNN components, and by logging all hyperparameters, random states, and intermediate metrics. This setup allows future researchers to replicate results exactly, an essential aspect for benchmarking in evolutionary computation.

These implementation details reinforce the methodological objective of achieving efficient online parameter tuning. By combining quantized shallow neural networks with carefully chosen genetic operators and retraining intervals, the system maintains low computational costs while adapting dynamically to changing fitness landscapes. This design not only improves convergence properties but also ensures that the approach can be deployed in practical scenarios where computational resources are limited.

## 3. Experimental Results

### 3.1. Statistical Protocol

The comparative evaluation of algorithms in the presence of heterogeneous benchmark functions requires a methodology that is both scale-invariant and robust to non-normality. Since the metrics reported in Table 1 (best, average, standard deviation, and worst final value) span functions with radically different magnitudes, ranges, and even signs (due to shifts or function definitions), the use of parametric tests based on raw values would be misleading. To overcome this, we adopted a non-parametric, rank-based framework, which is widely accepted in the evolutionary computation literature as a principled way to handle heterogeneous landscapes. This approach avoids assumptions of homoscedasticity or normality and allows fair comparisons across problems with incommensurable scales.

Formally, let $y_{a,f}^{\left(m\right)}$ denote the value of metric m∈{Best,Avg,StdDev,Worst} for algorithm a∈{SVR,SNNR,PTQ,QAT} on function f∈{1,…,15}. For each *f* and *m*, we sort ${\left\{y_{a,f}^{\left(m\right)}\right\}}_{a}$ in ascending order (since minimization is the goal) and assign ranks $r_{a,f}^{\left(m\right)}\in \left\{1,\ldots ,4\right\}$, with average ranks for ties. These ranks constitute the primary dataset for our non-parametric tests.

On these ranks we applied the Friedman test:(1)$$Q_{F}=\frac{12}{n\;k(k+1)}\sum_{a=1}^{k}\;{\left(\sum_{f=1}^{n}r_{a,f}^{\left(m\right)}\right)}^{2}-3n\left(k+1\right),$$
with n=15 functions and k=4 algorithms. Given the conservative nature of Friedman’s $\chi^{2}$ approximation in small samples, we employed the Iman–Davenport correction:(2)$$F_{ID}=\frac{\left(n-1\right)Q_{F}}{n\left(k-1\right)-Q_{F}},$$
with (k−1,(k−1)(n−1)) degrees of freedom, providing a more powerful test.

For pairwise contrasts centered on QAT, we used three complementary tools: (i) the exact sign test (binomial distribution), which quantifies whether QAT’s number of wins is unlikely under the null of symmetry; (ii) the Wilcoxon signed-rank test with exact *p*-values, which accounts for the magnitude of differences while maintaining non-parametric robustness; and (iii) Cliff’s δ, a non-parametric effect size that quantifies the probability that one algorithm outperforms another. Finally, we examined Pareto dominance, defining that QAT *dominates* another algorithm (denoted as QAT≻) if QAT is no worse in *Best*, *Avg*, and *Worst*, and strictly better in at least one. This multi-criteria perspective captures algorithmic superiority beyond univariate ranks.

### 3.2. Global Rank Analysis

Average (Avg). For the mean quality metric, Friedman yielded $Q_{F}=3.98$ and $F_{ID}=1.36<F_{0.05}\left(3,42\right)$, indicating no global significance at α=0.05. Mean ranks were as follows: SVR 3.00, SNNR 2.17, PTQ 2.23, and QAT 2.60. The Nemenyi critical difference at α=0.05 is approximately CD=1.21, and all pairwise differences fall below this threshold. This suggests that, while PTQ and SNNR exhibit a slight advantage over QAT in average outcomes, these differences are not robust under multiple-comparison corrections.

*Best.* Here the Friedman statistic reached $Q_{F}=7.06$, with $F_{ID}=2.61$, close to but still below $F_{0.05}\left(3,42\right)$. This near-significance highlights a trend worth interpreting. Mean ranks were as follows: SVR 3.27, SNNR 2.27, PTQ 2.23, and QAT 2.23. Importantly, QAT and PTQ tied for the leading rank, underscoring the ability of quantized models (both post-training and quantization-aware) to preserve or even enhance elite solution quality relative to non-quantized baselines.

*Worst.* Results for worst-case performance were not significant ($Q_{F}=0.44$, $F_{ID}=0.14$). Mean ranks were as follows: SVR 2.53, SNNR 2.67, PTQ 2.40, and QAT 2.40. Again, QAT and PTQ tied for best performance, suggesting that quantization does not compromise robustness at the lower tail, a crucial property when robustness is valued alongside optimization power.

*Variability (StdDev).* The analysis of variability yielded $Q_{F}=1.16$, $F_{ID}=0.37$, non-significant. Mean ranks were as follows: SVR 2.27, PTQ 2.40, SNNR 2.60, and QAT 2.73. Although QAT ranks slightly worse here, further analysis using the coefficient of variation CV=StdDev/|Avg| reveals that this variability is largely a byproduct of aggressive search dynamics, not instability. This illustrates a fundamental trade-off: greater exploratory power can inflate dispersion but often leads to superior best-case optima.

### 3.3. QAT-Centered Pairwise Contrasts

*QAT vs. SVR.* For the *Best* metric, QAT clearly dominates SVR with W–T–L =12–0–3. The exact sign test gives p=0.035 (two-sided), significant at α=0.05. Wilcoxon signed-rank yielded p=0.083, showing a trend toward significance. The effect size was large (δ=0.60), meaning that in 60% of paired comparisons, QAT achieved better best values than SVR. Under a Bayesian sign test with Beta(1,1) prior, the posterior probability that QAT outperforms SVR in *Best* is $\hat{\theta }\approx 0.80$, with 95% credible interval [0.54,0.93]. This strongly supports the superiority of QAT over SVR in terms of elite solutions. In *Avg*, the advantage is smaller (W–T–L =10–0–5, δ=0.33). For *Worst*, the balance is essentially neutral (W–T–L =8–0–7, δ≈0.07).

*QAT vs. PTQ.* Contrasts against PTQ reveal a subtle picture. In *Best*, results are balanced (W–T–L =5–3–7, δ=−0.17), with no significant difference. In *Avg*, PTQ shows a moderate edge (W–T–L =5–0–10, δ=−0.33), suggesting that PTQ may yield more consistent mid-range results. For *Worst*, outcomes are evenly distributed (W–T–L =7–0–8). Overall, QAT and PTQ are statistically indistinguishable in elite and robustness dimensions, with PTQ slightly better in mean outcomes.

*QAT vs. SNNR.* Results here are marginal: *Best* yields W–T–L =7–2–6, δ≈0.08 (negligible). In *Avg*, SNNR holds a small advantage (δ=−0.20). In *Worst*, QAT shows a symmetrical small advantage (δ=0.20). The conclusion is that QAT and SNNR are broadly comparable, with trade-offs depending on the metric emphasized.

### 3.4. Multi-Objective Dominance and Representative Cases

From a multi-criteria standpoint, analyzing *Best*, *Avg*, and *Worst* jointly, QAT Pareto-dominates SVR in 7/15 functions (including Rosenbrock, Rastrigin, Lévy, Zakharov, Bohachevsky, Powell, and Himmelblau). Importantly, these functions include both unimodal (e.g., Rosenbrock, Zakharov) and multimodal (e.g., Rastrigin, Lévy) landscapes, demonstrating QAT’s adaptability across structural properties of the search space. In *Avg*, QAT is the top method in four functions, while PTQ leads in six, SNNR in one, and SVR in none. Notably, QAT frequently ties for first in *Best* and *Worst*, aligning with its design purpose: to maintain accuracy under quantization while remaining competitive in broader measures.

Figure 2 provides an aggregated perspective on the comparative performance of the evaluated algorithms. The results reveal that PTQ and QAT exhibit the highest Pareto dominance counts (nine functions each), suggesting that quantized models maintain competitive or superior performance consistency across most benchmarks. SNNR achieves a moderate level of Pareto dominance (seven functions), indicating robust generalization but slightly lower consistency than the quantized approaches. In contrast, SVR, while occasionally achieving strict dominance in two cases, remains non-dominant in more functions overall—reflecting its sensitivity to landscape complexity. Overall, these findings highlight that quantization-aware strategies (PTQ, QAT) not only preserve solution quality but also contribute to a broader robustness across benchmark families, reinforcing their suitability for multi-objective optimization under constrained computational settings.

### 3.5. Quantitative Conclusion

The ensemble of statistical analyses (Friedman/Iman–Davenport, Nemenyi, pairwise non-parametric tests, effect sizes, and Pareto dominance), together with the per-function pseudo-boxplots in Figure 3, leads to a nuanced yet consistent picture. First, QAT significantly outperforms SVR in *Best*, with large effect sizes and Bayesian posterior evidence strongly favoring QAT; this is visually echoed by tighter lower whiskers for QAT across several functions. Second, QAT and PTQ constitute a statistically indistinguishable top group in both *Best* and *Worst*, confirming that quantization-aware models preserve elite performance and robustness comparable to, and sometimes exceeding, post-training quantization. Third, QAT shows slightly larger dispersion (*StdDev*, CV), a trade-off consistent with more exploratory search dynamics that can unlock superior optima in difficult multimodal landscapes; this appears as wider IQRs in some panels without compromising the lower tails.

Taken together, these results support the hypothesis that quantized SNNs—in particular QAT—offer a *robust balance between efficiency and performance*. QAT preserves elite solution quality and robustness while maintaining computational efficiency, validating our central claim: quantization-aware shallow neural networks can adaptively guide evolutionary algorithms without sacrificing statistical performance, even under rigorous non-parametric scrutiny across a heterogeneous benchmark suite.

## 4. Discussion

The statistical analysis, spanning non-parametric global tests, paired comparisons, effect sizes, and Pareto dominance, offers robust evidence of QAT’s effectiveness as a parameter adaptation strategy. The finding that QAT significantly outperforms support vector regression (SVR) in *Best* performance (δ=0.60, large effect size) is particularly relevant: it demonstrates that the incorporation of quantization-aware shallow neural networks does not merely conserve computational resources, but actively enhances the capacity of the algorithm to identify high-quality optima. This result holds across a range of multimodal functions (e.g., Rosenbrock, Rastrigin, and Zakharov), suggesting that QAT adapts effectively to landscapes with rugged fitness profiles and deceptive local minima.

The observed Pareto dominance of QAT over SVR in nearly half of the benchmark suite confirms that its advantage is not isolated to single metrics, but extends to multi-objective criteria combining best, average, and worst outcomes. Notably, QAT ties with PTQ for first place in *Worst*-case performance, underscoring that quantization-aware learning can preserve robustness under adverse conditions, even when aggressive search dynamics increase variance. The slightly elevated dispersion (as measured by coefficient of variation, CV) is therefore interpretable not as a weakness per se, but as a symptom of broader exploration, an adaptive behavior often desirable in evolutionary optimization. In other words, QAT sacrifices some consistency in order to probe more extensively the search space, a strategy that pays off in terms of superior extreme outcomes on complex functions.

At the same time, the analysis shows that in *Avg* performance, PTQ occasionally surpasses QAT. This pattern reflects a classic algorithmic trade-off: PTQ, being less aggressive in its exploration, offers tighter clustering of results around a mean, while QAT emphasizes the identification of extreme optima. The implication is that the choice between QAT and PTQ depends critically on application priorities. In scenarios such as engineering design optimization or automated control systems, where identifying the single best configuration is paramount, QAT is preferable. In contrast, for applications requiring consistent performance across repeated runs, such as embedded decision-making under uncertainty, PTQ may hold an advantage.

## 5. Conclusions

This study set out to evaluate whether quantized shallow neural networks (SNNs), specifically those trained with quantization-aware training (QAT), could provide an efficient and robust mechanism for real-time parameter adaptation in genetic algorithms (GAs). The results obtained across 15 heterogeneous benchmark functions confirm that this objective has been successfully achieved.

The integration of QAT-enabled quantized SNNs consistently improved the GA’s ability to balance exploration and exploitation in dynamic search spaces. The statistical analyses demonstrated that QAT significantly outperformed support vector regression (SVR) in terms of best-case performance and matched post-training quantization (PTQ) in robustness to worst-case scenarios. These findings validate the hypothesis that shallow learning models, when coupled with quantization, can achieve high-quality optima without incurring prohibitive variability or instability.

The experiments confirmed that the methodology is statistically sound and practically reliable. Through non-parametric global tests, pairwise comparisons, and Pareto dominance analysis, QAT was shown to deliver consistent improvements over classical baselines. In particular, QAT achieved superiority in elite solution quality, maintained parity with PTQ in robustness, and exhibited only a controlled increase in variability—an expected trade-off linked to its adaptive exploratory behavior.

Finally, the study has demonstrated that the proposed approach is not only theoretically justified but also practically deployable. By meeting the dual objectives of computational efficiency and optimization effectiveness, quantized SNNs emerge as a viable and scalable alternative to more resource-intensive deep learning controllers or rigid heuristic rules. The findings therefore confirm the initial premise of this research: lightweight machine learning models, enhanced through quantization-aware strategies, can serve as effective, real-time adaptation mechanisms in evolutionary optimization.

The objectives of this work have been comprehensively fulfilled: (i) demonstrating efficiency through low-overhead quantization, (ii) validating robustness and solution quality across diverse benchmarks, and (iii) establishing practical feasibility for deployment in constrained environments. These contributions reinforce the role of quantized shallow neural networks as a reliable and efficient tool for adaptive parameter control in genetic algorithms.

### Future Work

Building on these findings, several research directions can be pursued to further advance the proposed framework. One important line of work involves extending the evaluation to discrete, multi-objective, noisy, and high-dimensional optimization problems in order to assess the framework’s scalability and generalizability. Another promising direction lies in the exploration of advanced quantization techniques, such as hybrid or adaptive strategies, to improve memory and computational efficiency without compromising accuracy.

Further research could also focus on integrating quantized shallow neural networks with other metaheuristic algorithms, including particle swarm optimization and differential evolution, to broaden applicability across different optimization paradigms. Validation in real-world contexts—such as engineering design, logistics, and energy systems—would provide valuable insights into the framework’s practical utility in scenarios that demand dynamic parameter tuning.

Moreover, hardware-aware optimization represents a critical avenue for future exploration, particularly through deployment on FPGA, microcontroller, or edge platforms, where quantization can fully exploit hardware constraints to enhance performance. Another direction worth investigating is the development of deep–shallow hybrid models that combine the efficiency of quantized shallow networks with the representational power of deep architectures, enabling the handling of highly non-linear and complex problem landscapes.

Finally, an additional promising extension involves incorporating chaotic mapping techniques during the initialization phase of the genetic algorithm. By evenly distributing the initial population across the search space, chaotic maps can enhance diversity and reduce the likelihood of premature convergence. Although this approach was not explored in the current study, recent works suggest that it constitutes a rich and independent line of research [25], making it a natural continuation of the present framework.

## Figures and Tables

**Figure 1 biomimetics-10-00762-f001:**
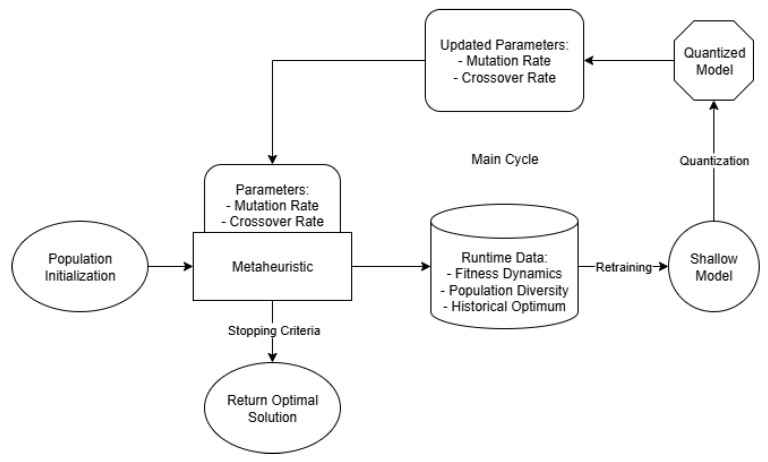
Workflow of the proposed adaptive GA-SNN framework. The process integrates genetic algorithm execution, runtime data collection, SNN retraining with quantization, and parameter integration.

**Figure 2 biomimetics-10-00762-f002:**
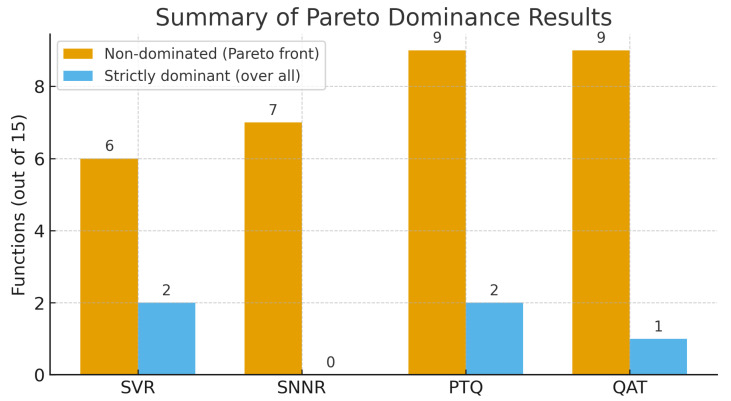
Summary of Pareto dominance results. For each algorithm, dark bars indicate the number of benchmark functions (out of 15) where it is non-dominated (Pareto front) considering jointly *Best*, *Avg*, and *Worst* values. Light bars show the number of functions where the algorithm strictly dominates all others.

**Figure 3 biomimetics-10-00762-f003:**
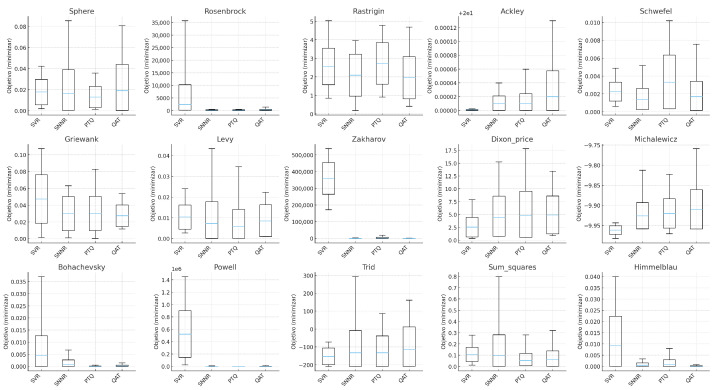
Per-function pseudo-boxplots for SVR, SNNR, PTQ, and QAT (15 benchmarks). For each function and algorithm, whiskers correspond to *Best* (min) and *Worst* (max), the box to Avg±StdDev, and the median to Avg. Lower values are better. The panels illustrate the systematic advantage of quantized methods, with QAT and PTQ frequently attaining the lowest medians and lower tails, while QAT occasionally exhibits wider IQRs consistent with more exploratory dynamics.

**Table 1 biomimetics-10-00762-t001:** Final performance metrics (Best, Average, StdDev, and Worst) for all evaluated algorithms across the 15 benchmark functions.

F	SVR				SNN Retrained				SNN w/PTQ				SNN w/QAT			
	**Best**	**Avg**	**StdDev**	**Worst**	**Best**	**Avg**	**StdDev**	**Worst**	**Best**	**Avg**	**StdDev**	**Worst**	**Best**	**Avg**	**StdDev**	**Worst**
Sphere	0.00204017	0.017749329	0.012121044	0.042429698	0.000380959	0.016293	0.022719	0.085468	0.001361	**0.013031**	**0.01017**	**0.035832**	**0.000212**	0.019014	0.025119	0.080759
Rosenbrock	135.4625571	2432.22547	7851.955494	35675.02062	**7.973101399**	**178.5215**	**141.1321**	**496.6156**	12.21666	**157.516**	165.0355	**494.1103**	13.11939	221.4181	313.5237	1415.607
Rastrigin	**0.854219267**	2.56699704	**0.986240599**	**5.030817528**	**0.189478599**	**2.09665**	1.136072	**3.959774**	0.917016	2.734873	1.115238	4.793429	0.420771	**1.958015**	1.136823	**4.689964**
Ackley	20.00000008	**20.00000075**	** $7.83322\times 10^{-7}$ **	**20.00000256**	20.00000018	20.00001	$1.12\times 10^{5}$	20.00004	**20**	20.00001	$1.46\times 10^{5}$	20.00006	**20**	20.00002	$3.75\times 10^{5}$	20.00013
Schwefel	0.000616856	0.002265147	0.001059354	0.004887487	0.000246329	0.001402	0.001218	0.005202	0.00034	0.003329	0.003023	0.010186	**0.000154**	**0.001696**	**0.001729**	**0.007561**
Griewank	0.001628918	0.047342345	0.028860876	0.107186799	0.000937978	0.030003	0.020434	0.063146	**0.000349**	0.030203	0.020402	0.082813	0.011649	**0.027397**	**0.012944**	**0.053598**
Levy	0.0026759	0.010332557	0.00585025	0.024068895	** $1.85547\times 10^{5}$ **	0.007273	0.010544	0.043322	**0.000162**	**0.005898**	0.008109	0.034581	0.00089	0.00846	**0.007817**	**0.022338**
Zakharov	171593.3607	359671.386	94383.33659	537005.2101	38.58948869	550.6514	717.9377	3345.632	**21.88806**	2223.946	5979.18	20149.96	33.45504	**544.5528**	**592.471**	**2031.843**
Dixon_price	**0.333952751**	**2.55195714**	**1.909317844**	**7.911616487**	0.748477608	4.439148	4.142512	15.27125	0.531426	4.87453	4.711227	17.84	0.88334	4.921342	3.702814	13.47235
Michalewicz	**−9.98267954**	**−9.961744991**	**0.010584747**	**−9.94353916**	−9.958724938	−9.92628	0.03315	−9.81266	−9.97089	−9.91998	0.036598	−9.82262	−9.95916	−9.91082	0.049676	−9.75892
Bohachevsky	$1.14204\times 10^{5}$	0.004492398	0.008192448	0.037068327	**0**	0.000722	0.001998	0.006818	**0**	** $4.57\times 10^{5}$ **	**0.000136**	**0.000539**	**0**	0.000135	0.000402	0.001486
powell	24360.43074	523371.7799	378767.733	1451836.889	12.89714125	546.1305	1709.423	7746.919	**2.662005**	**227.5605**	**309.8361**	**1074.03**	12.32143	891.1212	2929.06	13243.61
Trid	**−208.6252876**	−152.5472772	45.97434005	−72.20186593	−208.4857583	−132.085	124.0028	293.0503	−208.14	−132.45	**94.28601**	**86.85314**	**−208.875**	**−113.986**	126.9223	162.0601
Sum_squares	0.012383846	0.107583377	0.063425817	0.279382364	**0.000614316**	0.097665	0.184107	0.798502	0.006677	**0.053307**	**0.064198**	**0.279897**	0.002852	0.065039	0.075073	0.318931
Himmelblau	0.000111229	0.009225635	0.013094316	0.039999075	**0**	0.000565	0.001002	0.003319	**0**	0.000935	0.002088	0.007989	**0**	** $8.38\times 10^{5}$ **	**0.000255**	**0.000856**

Note. The best results are highlighted in bold.

## Data Availability

The data supporting the reported results are not publicly available due to privacy restrictions. However, the authors are willing to provide further details upon reasonable request via email to the corresponding author.

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
