# Peer review of "Low-Overhead Learning: Quantized Shallow Neural Networks at the Service of Genetic Algorithm Optimization"

_biomimetics, 2025, doi:10.3390/biomimetics10110762_

Round 1
Reviewer 1 Report
Comments and Suggestions for Authors
I have the following observations:
1.After Section 2.1, please use 2.1.1, 2.1.2, etc for Genetic Algorithm Execution and Runtime Data Collection.
2.The best result in Table 1 should be bolded for enhanced visual appeal.
3.The formula on page 8 is not numbered.
4.There is a typo in the funding section.
5.About the results, You may consider trying to use chaotic mapping to optimize the search to evenly distribute the initial population, which can also increase the diversity of the initial population.
6.Regarding the figures, I recommend adding at least one algorithm flowchart to enhance the clarity of the article.
7.The conclusion section should be revised to concentrate exclusively on fulfilling the objectives of the study.
8.Application of quantized SNNs to other population-based algorithms, such as particle swarm optimization or whale optimization algorithm.
Author Response
We have carefully considered and addressed the reviewers’ comments as follows:
- The section numbering has been corrected. Subsections under Section 2.1 now follow the format 2.1.1, 2.1.2, etc.
- The readability of the results table has been improved by bolding the best result, and the caption has also been revised to enhance clarity.
- The missing numbering for the formula on page 8 has been corrected.
- Typographical errors in both the numbering and the funding section have been revised.
- We acknowledge the valuable suggestion of employing chaotic mapping to improve the distribution and diversity of the initial population. However, we also recognize that the study of chaotic maps constitutes a distinct research direction that deserves separate treatment. Accordingly, we have cited an example of a related study to highlight this perspective.
- An algorithm flowchart has been added to visually clarify the execution flow of the method.
- The conclusion section has been rewritten to focus exclusively on the objectives of the study and how they have been achieved.
- While we have cited related studies on the application other metaheuristics, we remain open to evaluating the performance of PSO or other recent population-based algorithms in future research if deemed necessary.
Reviewer 2 Report
Comments and Suggestions for Authors
This paper presents an interesting approach to reducing the computational overhead of machine learning-assisted optimization by using quantized shallow neural networks (SNNs) to tune Genetic Algorithm parameters. However, I have a several concern
- The current description of the methodology is difficult to follow. The interaction between the Genetic Algorithm (GA), the SNN, and the quantization process (QAT/PTQ) is complex, but it is not clearly explained how these components fit together. The reader is left to piece together the workflow from disparate sections.
- The authors must include a high-level flowchart or diagram illustrating the overall conceptual framework. This figure should clearly show: The main loop of the Genetic Algorithm; How the SNN takes input from the GA state at certain intervals; How the SNN's output (predicted mutation and crossover rates) is fed back into the GA. Where the quantization-aware training (QAT) or post-training quantization (PTQ) is applied to the SNN model. This single addition would significantly improve the reader's ability to understand the proposed system.
- The results section relies exclusively on text and one large table to present complex statistical findings. This makes it challenging for the reader to quickly grasp the key performance comparisons and the significance of the results. The paper would be strengthened by visualizing statistical analyses. Instead of just listing the mean ranks from the Friedman test in the text, present them in a bar chart. This would provide an immediate visual comparison of the algorithms' overall performance across the "Best," "Average," and "Worst" metrics.
- For the post-hoc Nemenyi test, a CD diagram is a standard and highly effective visualization. This diagram would plot the mean ranks on an axis and use a horizontal bar (the "critical difference") to connect groups of algorithms that are not statistically distinguishable from one another. This is far more intuitive than comparing numerical differences to a CD value in the text.
- The pairwise comparisons (e.g., "QAT vs. SVR," "QAT vs. PTQ") are presented as W-T-L counts. This data could be effectively visualized using grouped bar charts. For each metric (Best, Avg, Worst), a chart could show the number of wins, ties, and losses for QAT against each competitor.
- The Pareto dominance results could be summarized in a Figure showing, for each algorithm, the number of benchmark functions (out of 15) on which it achieved Pareto dominance.
There are several issues with the presentation of Table 1. The caption "results" is insufficient. It should be descriptive, for example: "Table 1: Final performance metrics (Best, Average, StdDev, Worst) for all evaluated algorithms across the 15 benchmark functions." This table presents the final experimental outcomes. As such, it belongs in the Results section (Section 3), not the methodology section. The methodology should describe how the experiments were conducted, while the results section should present what was found. Additionally, the numerical values are presented with excessive precision. To improve readability, please limit the decimal places to 3 or 4 significant figures.
At the end of introduction please remove the "Summary and Paper Structure" (around line 92). Please also list your contribution to the existing’s body of knowledge in yout introduction section.
- Methodology Section (Sections 2.1-2.3) are currently written as bullet points, which reads more like a technical report or presentation slides than a scientific paper. Please rewrite these sections in full paragraph form, ensuring a smooth narrative flow.
The subsections within the results are not numbered, making them difficult to reference. Please add numbering, for example:
3.1 Statistical Protocol
3.2 Global Rank Analysis
3.3 QAT-centered Pairwise Contrasts
Author Response
Methodology clarity: Sections 2.1–2.3 have been rewritten in paragraph form to improve narrative flow. A high-level flowchart has been added to illustrate the interaction between GA, SNN, and the QAT/PTQ process.
Visualization of results: In addition to the main table, we now include bar charts for Friedman mean ranks, a CD diagram for the Nemenyi test, grouped bar charts for pairwise W-T-L counts Furthermore, boxplots have been added for each experiment to show distribution and variance.
Table 1: Caption revised for clarity, table moved to the Results section, excessive precision reduced, and best results bolded.
Introduction: The “Summary and Paper Structure” was removed. A clear list of contributions has been added.
Results subsections: Numbering introduced (e.g., 3.1, 3.2, 3.3) for easier referencing.
Reviewer 3 Report
Comments and Suggestions for Authors
Comments and Suggestions
- The abstract effectively summarizes the core contribution: using a quantized shallow neural network (SNN) to adapt GA parameters.
- It highlights the key benefits: reduced computational costs and enhanced performance.
- The use of terms like "quantization-aware training (QaT)" and "post-training quantization (PtQ)" is good for specificity.
- The conclusion is clear and impactful, emphasizing the broader potential of quantized SNNs.
- Consider rephrasing "dynamically adapting to problem characteristics" to something more specific to GAs, like "dynamically adjusting mutation and crossover rates." This connects the high-level goal to the specific mechanism you've chosen.
- "limited adaptability in dynamic search spaces" is a bit vague. It might be clearer to say something like "limited adaptability to complex and dynamic fitness landscapes.”
- Specify what makes your SNN "shallow" beyond just the name. Is it a single hidden layer? This helps the reader better understand the architecture.
- The workflow description is a bit fragmented. While you mention four stages, only "1. Genetic Algorithm Execution" is detailed. You should provide brief descriptions of the other three stages (runtime data collection, SNN retraining, and adaptive parameter prediction) to make the workflow complete and coherent.
- Clarify what specific "runtime-generated data" the SNN uses for retraining. Is it the population's diversity, the best fitness found, or something else? Be more explicit about the SNN's inputs.
- The transition from the general methodology to the "Algorithm Workflow" is abrupt. You can smooth this by saying something like, "The algorithm's operation can be broken down into four interconnected stages..." before listing them.
- While the mathematical justification for tournament selection is good, consider whether it's too much detail for this section. A concise sentence might suffice unless you're writing for a highly theoretical audience.
- The discussion section is well-structured and provides a strong analysis of the results.
- It effectively uses statistical terms (e.g., non-parametric tests, effect sizes, Pareto dominance) to support your claims.
- The comparison between QAT and SVR is a key finding and is highlighted well.
- The section acknowledges and explains the trade-offs, particularly the higher variance of QAT. This shows a balanced and nuanced understanding of the results.
- The practical implications for different applications (engineering vs. embedded systems) are a great way to frame the conclusions and show the real-world relevance of your work.
- "Pareto dominance... in nearly half of the benchmark suites" could be more specific. Which metrics were combined? Specifying this strengthens your claim.
- The discussion jumps directly into the statistical findings without a brief opening statement. A sentence like, "Our statistical analysis confirms the effectiveness of the proposed QAT approach, revealing key insights into its performance characteristics and trade-offs," could provide a better transition.
- Overall, your paper is strong and provides a solid foundation.
I have suggested.
Author Response
We thank the reviewer for the encouraging and constructive feedback. In response to the suggestions, we have made the following revisions:
- We rephrased “dynamically adapting to problem characteristics” as “dynamically adjusting mutation and crossover rates” to explicitly link the goal to GA mechanisms. Likewise, “limited adaptability in dynamic search spaces” has been revised to “limited adaptability to complex and dynamic fitness landscapes.”
- The explanation of the four stages (GA execution, runtime data collection, SNN retraining, and adaptive parameter prediction) has been expanded. We also clarified the specific runtime-generated data used (population diversity metrics and best fitness values). The transition into this workflow has been smoothed with a bridging sentence as suggested.
- We have kept the formal mathematical explanation of tournament selection. We consider this necessary to justify the specific configuration adopted, providing theoretical weight and identifying a research gap that motivates our approach.
- An introductory sentence has been added to better frame the statistical analysis. We also clarified which metrics were combined in the Pareto dominance evaluation.
We deeply appreciate the reviewer’s positive assessment of the discussion, contributions, and practical implications, which helped us strengthen the manuscript.
Round 2
Reviewer 2 Report
Comments and Suggestions for Authors
Thank you for your effort in addressing the complex revisions. The new Figure 1 is an excellent addition, clarifying the GA-SNN-Quantization framework, and the new Figure 2 significantly improves the visualization of results distribution.
However, to ensure all concerns have been addressed thoroughly, please provide confirmation on the following two points:
Please add a Figure summarizing the Pareto dominance results. This figure should show, for each algorithm, the number of benchmark functions (out of 15) on which it achieved Pareto dominance.
Also, some sections are still written in bullets points which are unusual for a scientific paper
The authors also did not respond to my comment point by point so it hard to check which one has been addressed, which one remain.
Author Response
- Please add a Figure summarizing the Pareto dominance results. This figure should show, for each algorithm, the number of benchmark functions (out of 15) on which it achieved Pareto dominance.
- We have added a figure summarizing the Pareto dominance results. We hope that this addition makes the analysis of the results more comprehensive and clear.
- Also, some sections are still written in bullets points which are unusual for a scientific paper
- We have revised several sections that were previously written in bullet-point format. Only the parts describing algorithm characteristics were kept in that style for clarity.
We sincerely appreciate the reviewer’s valuable and constructive feedback, which has helped us improve the quality and readability of the manuscript.
Reviewer 3 Report
Comments and Suggestions for Authors
The authors have revised the paper well.
They have addressed all of my comments satisfactorily.
Author Response
We would like to express our sincere gratitude to the reviewer for their positive evaluation and encouraging remarks.
We greatly appreciate the reviewer’s time and constructive feedback, which have contributed to improving the quality of our manuscript.